# Current Strategies to Target Tumor-Associated-Macrophages to Improve Anti-Tumor Immune Responses

**DOI:** 10.3390/cells9010046

**Published:** 2019-12-23

**Authors:** Clément Anfray, Aldo Ummarino, Fernando Torres Andón, Paola Allavena

**Affiliations:** 1IRCCS Istituto Clinico Humanitas, Via A. Manzoni 56, 20089 Rozzano, Milan, Italy; clement.anfray@humanitasresearch.it (C.A.); fernando.torres.andon@usc.es (F.T.A.); 2Humanitas University, Via Rita Levi Montalcini 4, 20090 Pieve Emanuele, Milan, Italy; aldo.ummarino@hunimed.eu; 3Center for Research in Molecular Medicine & Chronic Diseases (CIMUS), Universidade, de Santiago de Compostela, Campus Vida, 15706 Santiago de Compostela, Spain

**Keywords:** tumor-associated macrophages, immune system, tumor microenvironment, immune suppression, cancer immunotherapy, clinical trials

## Abstract

Established evidence demonstrates that tumor-infiltrating myeloid cells promote rather than stop-cancer progression. Tumor-associated macrophages (TAMs) are abundantly present at tumor sites, and here they support cancer proliferation and distant spreading, as well as contribute to an immune-suppressive milieu. Their pro-tumor activities hamper the response of cancer patients to conventional therapies, such as chemotherapy or radiotherapy, and also to immunotherapies based on checkpoint inhibition. Active research frontlines of the last years have investigated novel therapeutic strategies aimed at depleting TAMs and/or at reprogramming their tumor-promoting effects, with the goal of re-establishing a favorable immunological anti-tumor response within the tumor tissue. In recent years, numerous clinical trials have included pharmacological strategies to target TAMs alone or in combination with other therapies. This review summarizes the past and current knowledge available on experimental tumor models and human clinical studies targeting TAMs for cancer treatment.

## 1. Introduction

Macrophages are innate immune cells belonging to the mononuclear phagocyte system, which include macrophages resident in peripheral tissues and circulating monocytes newly recruited at sites of inflammation and tissue damage (e.g., tumors). A key feature of macrophages is their phenotypical and functional plasticity, usually defined as polarization, which is dictated by their continuous adaptation and response to specific local stimuli. For instance, macrophages can act as pro-inflammatory and immune-stimulatory effectors in the defense against pathogens, or as anti-inflammatory cells devoted to the healing and remodeling of injured tissues [1,2,3].

To fulfill such different immune functions, macrophages acquire specific phenotypes that can be characterized in terms of gene expression, the pattern of surface molecules, and the production of biological mediators and metabolites [4,5,6]. At the edges of the continuum polarization status of macrophages, two extreme phenotypes can be defined as M1 pro-inflammatory/anti-tumor versus M2 anti-inflammatory/pro-tumor. M1-like macrophages, activated by lipopolysaccharides (LPS) and pro-inflammatory cytokines, such as IFNγ, present the ability to kill tumor cells, inhibit angiogenesis, and promote adaptive immune responses [3,5]. However, the uncontrolled activation of inflammatory M1 macrophages could represent a risk for the organism. Thus, over time inflammatory macrophages typically shift towards an M2 polarization. At the other extreme, M2-like macrophages, which mimic tumor-associated macrophages (TAMs) present in the tumor microenvironment (TME), can be induced by anti-inflammatory cytokines, such as IL-4 or IL-13. It has been experimentally demonstrated that TAMs or M2-like macrophages promote tumor initiation, progression, and survival; they inhibit immune-stimulatory signals and are devoid of cytotoxic activity [3]. TAM infiltration in tumors has been correlated with poor prognosis [3]. Furthermore, numerous investigations have revealed that TAMs are primarily responsible for resistance to classical anti-tumor treatments (i.e., chemotherapy or radiotherapy), and they also limit the efficacy of new immunotherapies (i.e., anti-PD1) [3,7,8,9]. These findings called attention to TAMs as promising targets of novel anti-tumor therapeutic approaches.

In this review, we provide an overview of the recent investigations related to TAM interaction with current clinical treatments, which limits their anti-tumor efficacy. In addition, pre-clinical experimentation and clinical trials using TAM-targeted strategies, alone or in combination with chemotherapies, checkpoint blockade immunotherapy, targeted therapy, or radiotherapy, are presented with the aim to provide an overview of the potential of macrophage-targeting approaches for the treatment of cancer.

## 2. Origin and Role of Macrophages in Cancer

In solid tumors, macrophages can represent up to 50% of the mass, becoming the main immune population. TAMs originate mostly from circulating precursor monocytes, but resident macrophages can be originally present in the tissue, later developing in a tumor [10,11]. The origin of monocytes in adults is related to a common myeloid progenitor, which depends on M-CSF (CSF-1) to differentiate into macrophages. Inflammatory monocytes are rapidly recruited at sites of tumor growth, following specific signaling by chemokines (e.g., CCL2), but also CSF-1, cytokines, or complement components (C5a) [3]. Resident macrophages, instead, originate from embryonic precursors that have migrated at peripheral tissues early in life [6].

The origin of TAMs within the tumor (resident macrophages vs. circulating monocytes) is not a mere classification connected to their localization but seems to influence their activity and phenotypic profile [12,13]. Indeed, Franklin and Li showed in murine models of breast cancer that depletion of resident TAMs did not influence tumor growth, while the absence of tumor-recruited TAMs (originating from circulating precursors) resulted in a better outcome [14]. In some tumors, the origin of TAMs is controversial: For example, in brain neoplasia, probably because of the presence of the blood–brain barrier, most (but not all) TAMs derive from resident microglia rather than circulating monocytes [15].

### 2.1. Tumor Microenvironment and Its Relation with TAMs

The relationship between cancer cells, macrophages, and other components of the TME is dynamic and heterogeneous. Considering the evolution of tumors, this immuno-suppressive and pro-angiogenic micro-environment is the physiological result of a process of prolonged inflammation and continuous tissue damage and remodeling. Tumor cells and immune cells in the TME produce cytokines, growth factors, and metabolites, which promote the pro-tumor polarization of TAMs. Biological mediators, such as CSF-1, CCL2, and vascular endothelial growth factor (VEGF), promote the accumulation of TAMs in the TME [4,16,17,18,19]. The Th2 cytokines IL-4, IL-13, IL-10, and TGFβ produced by Treg and TAMs are key drivers of immune-suppression [5,20].

Acidification of the TME caused by lactate derived from enhanced glycolytic activity of cancer cells induces regulatory macrophages through G protein-couple receptor (GPCR) and IL-1 beta-converting enzyme (ICE) [21], enhances VEGF and arginase expression, thus promoting M2-like features of TAMs [22].

The hypoxia-inducible factor 1 (HIF-α) is a master transcriptional regulator of cellular response to low oxygen concentration [23,24,25]. Indeed, the ability of different cells to sense and adapt to oxygen availability has been recognized by the Nobel Prize in Physiology and Medicine in 2019 [26], and this also applies to TAMs. Especially in advanced tumors, TAMs accumulate in hypoxic areas; these TAMs are MHC^low^, have pro-angiogenic behavior and poor antigen-presenting ability; on the other hand, macrophages localized in areas of normoxia, may be more heterogeneous, and some of them may present an M1 orientation with MHC^high^ expression [23,27]. Wenes et al. have shown that hypoxic TAMs upregulate REDD1, and endogenous inhibitor of MTORC1, leading to a decrease in glucose intake by TAMs and to higher availability for endothelial cells, thus promoting neo-angiogenesis and metastasis [28].

### 2.2. Functions of TAMs in the Tumor Microenvironment

Although topographical and temporal heterogeneity occurs in tumors, macrophages typically follow a continuum of pro-tumor activation states which can be summarized as follows:Production of growth factors for tumor cells;Promotion of neo-angiogenesis;Release of proteases and other molecules to remodel the extracellular matrix;Secretion of immuno-suppressive mediators (e.g., for T cells), which sabotage the ability of the host immune system to fight cancer.

The epidermal growth factor (EGF) secreted by TAMs promotes the proliferation and invasion of cancer cells, while VEGF regulates angiogenesis. In addition, VEGF receptors on the surface of TAMs signal an autocrine loop, which reinforces their pro-angiogenic and immuno-suppressive functions [3,28,29]. In a similar manner, Tie-2-expressing macrophages bind angiopoietins (Ang-1, Ang-2, etc.) and promote angiogenesis [30]. TAMs also release pro-angiogenic chemokines (e.g., CXCL8) and proteolytic enzymes, such as MMPs and cathepsins, which degrade the extracellular matrix (ECM) inducing the release of angiogenic factors previously stored in the ECM in an inactive form. As an example, it has been demonstrated that MMP-2 and MMP-9 activate TGFβ, VEGF, PDGF, and FGF, and their expression has been correlated to increased tumor invasiveness and worse prognosis [3,29,31]. A similar mechanism has been described for the urokinase-like plasminogen activator (uPA), which, upon binding to its receptor (uPAR) on TAMs, induces the cleavage of plasminogen into plasmin, resulting in ECM degradation and consequent release of growth factors and poor outcome [32].

The key driver of metastasis is TGFβ, which can be produced by TAMs; TGFβ triggers the epithelial-to-mesenchymal transition (EMT), shifting epithelial cancer cells to gain a mesenchymal phenotype, more convenient for motility [33]. The immuno-suppressive function of TAMs is also mediated by the secretion of TGFβ, as well as IL-10, that suppress CD8+ T cell functions by direct transcriptional repression of genes encoding functional mediators, such as perforins, granzymes, and cytotoxins; but also indirectly by stimulating the amplification of Treg cells or by suppressing DC anti-tumor functions [20,34,35]. Moreover, TAMs induce amino acid metabolic starvation in T cells through the production of arginase and indoleamine 2,3-dioxygenase (IDO) [25].

In response to hypoxia and to specific cytokines, TAMs overexpress the PD-1 ligands (PD-L1 and PD-L2), as well as the CTLA-4 ligand [36,37]. High PD-L1 expression in TAMs has been reported in different types of cancer, such as hepatocellular carcinoma [38], glioblastoma [39], or pancreatic cancer [40,41]. The lack of stimulatory signals from TAMs towards Th1 or CD8+ cytotoxic T cells also contributes to the anergic state of the adaptive immune system in the TME [3]. Finally, TAMs produce some pro-inflammatory cytokines (i.e., IL-6, TNFα), which contribute to the “smoldering inflammation” present in the TME, which in the long run causes a state of immunosuppression.

Figure 1 synthesizes the pro-tumor and potential anti-tumor functions of TAMs.

## 3. TAMs Hamper the Efficacy of Current Treatments in Clinical Oncology

TAMs interfere with most of the anti-tumor therapies commonly used in clinical practice: conventional chemotherapy, anti-angiogenic, radiotherapy, and the novel antibody-based immunotherapies targeting the molecules PD-1/PD-1L and CTLA-4: immune checkpoint blockade (ICB) [3,8].

### 3.1. Interaction of TAMs with Immune-Checkpoint Blockade Therapy

Immune checkpoints are represented by a family of proteins on the surface of T cells, which interact with specific ligands on antigen-presenting cells or cancer cells, and inhibit their TCR-mediated activation. In the last 5 years, anti-checkpoint antibodies have become the holy grail of tumor immunotherapy, resulting in outstanding clinical responses in selected cancer types (i.e., melanoma, lung, and renal cancer). Unfortunately, clinical efficacy is obtained only in a proportion of the treated patients. Moreover, certain cancer types, including pancreatic, colorectal, and ovarian cancer, are refractory to ICB therapy or show little benefit [9,42].

The ability of TAMs to limit the efficacy of ICB therapy has been demonstrated [43,44]. TAMs express ligand molecules for checkpoint receptors, such as PD-L1/2, CD80, CD86 and VISTA (V-domain immunoglobulin suppressor of T cell activation), and possibly others; the presence of checkpoint inhibitors different from those targeted by the currently available antibodies cancels the benefit of the therapy and maintains a state of strong immunosuppression [40,45,46,47]. The expression of PD-L1 by TAMs results in the sequestration of anti-PD-L1 mAbs [45]. Similarly, TAMs can bind anti-PD-1 with their Fc receptor; using intra-vital microscopy, Arlauckas et al. demonstrated that anti-PD-1 co-localizes with T cells for just a short period (0.5h), then the antibody was captured by TAMs for longer time (20 h), de facto hampering the efficacy of the ICB [43].

On the other hand, it is well recognized that macrophages can actively contribute to the clinical efficacy of therapeutic mAbs, such as Rituximab, designed to target B cells by killing the target via a mechanism of antibody-dependent cellular cytotoxicity (ADCC) [48]. This could be of advantage in the case of anti-PD-L1 targeting cancer cells but could be deleterious in the case of anti-PD-1 mAbs killing PD1+ T lymphocytes.

Several investigations have demonstrated that activation of PI3Kγ signaling in macrophages inhibits NF-κB, thereby promoting immune suppression. In several tumor models, the pharmacological inhibition of PI3Kγ or the stimulation of NF-κB was able to synergize with ICB therapy to promote tumor regression [49,50,51,52]. Other studies showed that high expression by TAMs of the leukemia inhibitory factor (LIF), a member of the IL-6 family, triggers the epigenetic silencing of CXCL9, a chemo-attractant for CD8+ T cells, and LIF inhibition improved the efficacy of anti-PD-1 therapy [53]. Finally, several studies have described how TAMs also interfere with the correct recruitment and localization of CD8+ T cells within the tumor, thus precluding the efficacy of ICB therapy [54,55].

### 3.2. Interaction of TAMs with Chemotherapy

TAMs in tumors treated with chemotherapeutic agents present dual effects: occasionally improving treatment efficacy, but more frequently being primarily responsible for chemoresistance. Among positive effects of TAMs on chemotherapy, early studies by Mantovani pointed out that host defense mechanisms played by macrophages contributed to the therapeutic efficacy of doxorubicin, a concept that has been expanded by Zitvogel’s group, as the ability of selected drugs to activate an immunogenic cell death (ICDs) that stimulates anti-tumor immune responses [56,57]. Cancer cells treated with chemotherapy are more susceptible to the cytotoxic effect of macrophages [58], and specific drugs, such as gemcitabine, have been reported to stimulate their cytotoxic potential and M1-like differentiation [59]. On the other hand, TAMs hamper the efficacy of chemotherapeutic drugs by the following mechanisms: (i) increased recruitment of immuno-suppressive myeloid cells, (ii) suppression of adaptive anti-tumor immune responses, (iii) activation of anti-apoptotic programs in cancer cells.

Commonly, chemotherapy-induced tissue damage promotes the recruitment of immuno-suppressive myeloid cells through the secretion of IL-34 and CSF-1 from the cancer cells, in the attempt to heal the injured tissues [60]. In the case of ICD activators, such as doxorubicin, controversial results have been reported, depending on the tumor type and the response of the microenvironment. Breast cancers treated with doxorubicin showed promotion of CCL2 production by stromal cells and consequent recruitment of CCR2+ monocytes, which contributed to tumor relapse [61]. Similarly, in colorectal cancer, 5-fluorouracil (5-FU) increased TAM recruitment in clodronate-depleted tumors [62]. In the same study, Zhang et al. showed that conditioned media from 5-FU-treated macrophages injected in mouse tumors resulted in the inhibition of 5-FU anti-tumor efficacy, being polyamine putrescine, an important component involved in this chemoresistance [62]. In a transgenic mouse model of breast cancer, Salvagno et al. found resistance to platinum-based therapy via the downregulation of type I IFN-stimulated genes (ISGs) in TAMs, which was reverted by CSF-1R blockade therapy and resulted in improved cisplatin efficacy [63]. Platinum-therapy induces the release of fatty acids that act on F4/80^+^/CD11^low^ macrophages in the spleen, promoting the release of polyunsaturated lisophosphatidylcholines that alter the DNA damage response and cause resistance to treatment [64]. As TAMs modulate the infiltration and activity of other immune cells through the production of chemokines or metabolites, any modification induced by chemotherapy on macrophages may interfere with the anti-tumor activity. As an example, Ruffel et al. showed that paclitaxel or carboplatin treatment, in mice bearing MMTV-PyMT tumors, is counteracted by the increased secretion of IL-10 by TAMs, which downmodulates IL-12 production in DCs and inhibits CD8+ T cell anti-tumor activity [65].

### 3.3. Interaction of TAMs with Anti-Angiogenic Therapy

TAMs are important mediators of the angiogenic switch in tumors and produce growth factors and other molecules which promote the vessel network, therefore, interfering with anti-angiogenic drugs. In parallel, the pharmacological blocking of angiogenesis, for example, with anti-VEGF mAbs, can cause blood vessel damage, which induces hypoxia and stimulates the secretion of myeloid cell chemoattractants [30]. The comparison of resistant versus sensitive tumors to anti-VEGF therapy revealed a higher number of TAMs in the non-responders [66]. In the same line, anti-VEGF mAbs showed higher therapeutic efficacy and higher inhibition of vessel formation, upon depletion of TAMs with zoledronic acid, in several tumor models [67]. This effect was also observed in murine glioblastoma treated with vatalanib, a small protein kinase inhibitor that blocks angiogenesis. The treatment increased TAM infiltration, and the beneficial effect of the anti-angiogenic therapy was significantly improved with the co-administration of anti-CSF-1R mAb, that impaired TAM recruitment [68]. Finally, the administration of a fully-humanized mAb blocking the Ang2-Tie2 interaction in Tie2-expressing macrophages resulted in the reduction of angiogenesis, but enhanced the recruitment of macrophages, in murine models of breast and pancreatic cancer [30]. TAM reprogramming and depletion of Tie2-expressing macrophages were combined with anti-VEGF therapy for enhanced anti-glioma responses [69].

### 3.4. Interaction of TAMs with Radiotherapy

Radiation therapy (RT) is a widely used and highly cost-effective cancer treatment modality. Interestingly, controversial results have been observed for macrophages in tumors exposed to RT. These could be explained by the former use of conventional RT versus new advanced equipment allowing for precise control of the dose, time, and localization of the RT-treatment, which could be crucial to achieving stimulation or inhibition of immune responses. For example, glioblastoma tumors treated with X-ray radiation showed a general decrease in macrophages and increased M2/M1 ratio of TAMs, and this effect was explained in vitro by the higher resistance of M2 versus M1 macrophages to radiation [70]. Others showed that macrophages irradiated with conventional RT sustain cancer cell invasion and angiogenesis [71], and these observations could be supported by experimental studies demonstrating the improved outcomes of RT in tumor-bearing mice pre-treated with clodronate liposomes to deplete macrophages [72].

On the other hand, several studies have demonstrated that low doses of RT reprogram macrophages towards an iNOS+/M1 phenotype [73,74].

This pro-inflammatory effect can be explained by the ability of RT to “destroy” cancer cells in a similar way to ICD activators, resulting in the release of “danger” signals (DAMPs), such as dsRNA or tumor antigens, which can be recognized by the patient’s immune system, triggering effective anti-tumor immune responses [71].

## 4. Targeting TAMs: Pre-Clinical Experimentation and Clinical Trials

As described before, TAMs are involved in many processes that promote tumor growth, including angiogenesis, invasion, metastasis, immunosuppression, and resistance to therapies [3]. The understanding of the molecular mechanisms implicated in these processes and the characterization of TAMs in tumors allowed for the development of original anti-tumor therapeutic strategies, and some of them are currently under evaluation in clinical trials. As a whole, TAM targeting strategies can be divided into two groups: i) strategies designed to inhibit TAM infiltration in tumors, either through inhibition of recruitment or direct killing, and ii) strategies aimed at reprogramming their pro-tumorigenic polarization to activate their anti-tumor functions.

Figure 2 summarizes the available strategies to target TAMs to improve anti-tumor immune responses.

### 4.1. Strategies to Inhibit the Recruitment or Number of TAMs in Tumors

#### 4.1.1. Strategies to Inhibit TAMs Recruitment

Strong evidence indicates that the accumulation of macrophages in tumors is due to the continuous recruitment of monocytes from the circulation in response to tumor-derived factors (TDFs). These TDFs are key mediators in the crosstalk between monocytes and tumor cells and include colony-stimulating factor-1 (CSF-1), several C−C chemokine ligands, such as CCL2, also known as MCP-1 and VEGF [75,76]. CCL2 has been described as the major TDF involved in monocyte recruitment, through the CCL2-CCR2 axis. Hence, blockade of CCR2 can suppress the accumulation of TAMs in tumors [75]. CCR2 inhibitors and anti-CCL2 antibodies have shown efficacy in reducing tumor growth and metastasis in several pre-clinical murine models [76]. In addition, anti-CCL2 antibodies were able to improve the efficacy of chemotherapy when administered simultaneously [77]. However, contrasting results were observed in other pre-clinical models. For example, in murine breast cancer, a rebound effect has been noted after the withdrawal of anti-CCL2 treatment, which has been associated with increased mobilization and infiltration of bone-marrow monocytes in the tumor and consequent acceleration of lung metastasis [78]. Clinical trials using anti-CCL2 antibodies have been performed in patients with prostate cancer (Table 1). CCR2 antagonists are being studied as monotherapy in patients with metastatic cancers, or in combination with chemotherapy (FOLFIRINOX) in advanced pancreatic adenocarcinomas, overall with limited results. Anti-CCL2 antibodies were effective in overcoming resistance to radiotherapy in murine models of pancreatic tumors [72].

Another combination therapy, investigated in ongoing Phase Ib/II clinical trials, consists in the use of a dual CCR2/CCR5 antagonist plus chemotherapy or nivolumab in patients with metastatic colorectal and pancreatic cancer. CCR5 has been related to the recruitment of polymorphonuclear MDSCs (PMN-MDSCs) and T cells in the TME. Thus, it will be of interest to evaluate the balance efficacy/safety of CCR2- or CCR5-selective inhibition versus dual inhibition in combination with chemotherapy or anti-PD-1 therapy (Table 1).

Another axis involved in monocyte recruitment and differentiation into TAMs is the CXCL12/CXCR4 axis [79,80]. In a breast cancer model, expression of CXCL12 by tumor cells increased macrophage and vessel density, contributing to the invasion ability of tumor cells. Inhibition of CXCR4 with the antagonist AMD3100 reduced tumor cell spreading and formation of metastasis [81]. AMD3100 is under evaluation in combination with pembrolizumab, in a recently initiated clinical trial for patients with refractory head and neck squamous cell carcinoma (NCT04058145). Previously, a couple of clinical trials investigated the safety of continuous intravenous administration of another CXCR4 antagonist (plerixafor) and its impact on the tumor microenvironment (NCT02179970, NCT03277209) in patients with solid tumors. Plerixaflor is also under evaluation in patients with acute myeloid leukemia (AML) in combination with G-CSF, which downregulates CXCL12 expression and acts synergistically in stem cell mobilization (NCT00906945), or in combination with several chemotherapeutics to treat relapsed or refractory AML (NCT01236144, NCT01220375, NCT01027923). Similar studies are being performed with other CXCR4 antagonists (NCT01010880, NCT02907099, NCT02115672, NCT02954653, NCT02737072, NCT01359657, NCT01120457).

The inhibition of CXCL12 reduced the myeloma-supportive activity of the bone marrow microenvironment and mobilized myeloma cells into the circulation. Olaptesed pegol (NOX-A12) is a pegylated L-oligoribonucleotide that binds and neutralizes CXCL12, achieving the mobilization of myeloma cells for at least 72 h and enhancing the activity of dexamethasone without relevant additional toxicity [82] (NCT04121455). This approach is also being investigated in combination with pembrolizumab in patients with colorectal and pancreatic cancer (NCT03168139).

#### 4.1.2. Strategies to Deplete TAMs

The growth factor CSF-1 is involved in the proliferation, differentiation, and survival of monocytes/macrophages that originated from bone marrow progenitor cells. A high level of CSF-1 or CSF-1R expression in the tumor or peri-tumor tissue has been associated with poor patient survival in lymphoma, breast cancer, and hepatocellular carcinoma [83,84,85,86]. Strong evidence supports the CSF-1/CSF-1R axis as an attractive target to reduce the number of TAMs in tumors. Consequently, the pharmacological inhibition of the CSF-1/CSF-1R axis alone or in combination with different therapies has been explored in pre-clinical settings and in clinical trials, preferentially in patients with advanced solid tumors. Antibodies and small molecules have been developed to target the CSF-1 receptor or its downstream signaling pathways (Table 1). In animal models, emactuzumab, a humanized mAb targeting CSF-1R, resulted in decreased TAM numbers in the tumor and increased CD8+/CD4+ T cell ratio [87]. The evaluation of this mAb alone or in combination with paclitaxel in patients with advanced solid tumors revealed an important reduction of TAMs in the TME and a good safety profile, but so far, no clinically relevant improved outcomes [88,89]. Similar mAbs targeting CSF-1R are currently being investigated in clinical trials as monotherapy or in combination with ICB for advanced solid tumors and with cyclophosphamide in patients with ovarian cancer (Table 1).

Macrophage depletion by CSF-1R blockade with small molecule inhibitors also showed increased infiltration of CD8+ cytotoxic T cells in the tumor and improved response to therapies in murine models of breast, prostate, and cervical tumors [60,90,91]. TAMs can trap CD8+ T cells at the tumor periphery, thus having a detrimental impact on T cell motility inside the tumor. The inhibition of CSF-1R using the small drug PLX3397 was effective to deplete TAMs and to restore T cell migration into the TME of mouse mammary tumor models [55]. In pancreatic cancer, not responding to ICB therapy, a combination of anti-PD1 or anti-CTLA-4 checkpoint immunotherapy with PLX3397 improved anti-tumor immunity and led to the regression of established primary pancreatic tumors [92]. PLX3397 has also been combined with the TORC1 inhibitor rapamycin for the treatment of malignant peripheral nerve sheath tumors. This highly aggressive tumor, resistant to chemotherapy and to imatinib, showed a significant depletion of macrophages and tumor reduction after PLX3397 treatment, which was improved by its combination with rapamycin, even when drug treatment was discontinued [93].

Inhibition of the CSF-1/CSF-1R axis is presently explored in Phase I/II clinical trials in patients (Table 1). The combination PLX3397 plus eribulin, a macrocyclic ketone analog that inhibits microtubule dynamics, is being tested in patients with metastatic breast cancer and soft tissue sarcoma. PLX3397 also improved the efficacy of the BRAF inhibitor veramufenib in murine models of melanoma, and this therapeutic combination is ongoing in patients with BRAF^V600^ mutant metastatic melanoma [94]. Other recent studies suggested that inhibition of the CSF-1/CSF-1R axis is a promising strategy to treat primary and metastatic KRAS-tumors resistant to immune checkpoint blockade. As an example, CSF-1/CSF-1R blockade decreased fibrosis of primary and metastatic tumors in KRAS pancreatic mouse models, resulting in increased CD8+ T cell infiltration [41,95]. Other studies have reported on the ability of CSF-1R inhibition alone or in combination with other therapies to induce TAM reprogramming towards an anti-tumor M1 phenotype [96] (see Section 4.2.4).

Radiotherapy has been combined with several TAM-targeted therapies. Of note, instead of depleting TAMs, the inhibition of CSF-1 after ionizing radiation resulted in altered myeloid cell recruitment and polarization. PLX3397 plus radiotherapy increased CSF-1 expression and myeloid cell infiltration in preclinical mouse xenograft models of human glioblastomas [97]. Clinical trials are currently studying the combination of PLX3397 and radiotherapy with temozolomide for glioblastoma, or androgen deprivation therapy for prostate cancer (Table 1). Combinations of anti-CCL2 antibodies with radiotherapy are also being tested in pre-clinical and clinical studies [72,98].

Previous to the discovery of compounds targeting the CSF-1/CSF-1R axis, other drugs with preferential cytotoxic activity towards monocytes/macrophages have been used. For example, bisphosphonates, which are typically used for the treatment of osteoporosis and for the prevention of complications associated with bone metastases. Following administration, these inorganic compounds are selectively adsorbed into the bone tissue and metabolized by osteoclasts, thus limiting systemic exposure [29]. In addition to osteoclasts, tissue macrophages, including TAMs, have been reported to be affected by bisphosphonates, alone or in combination with chemotherapy. Several nanoparticles (NPs) have been used to improve the delivery of bisphosphonates to TAMs, the most popular being liposomal formulations of clodronate, which showed anti-tumor efficacy in several murine models of colon cancer, glioma, and myeloma [3,29]. Clodronate liposomes are commonly used in biomedical research to deplete macrophages to understand their function in vivo. However, the high toxicity of this approach limits the translational relevance of these studies [99,100].

Trabectedin is a registered anti-neoplastic drug, which in addition to targeting tumor cells, can partially deplete circulating monocytes and TAMs through a TRAIL-dependent pathway of apoptosis [101]. Our group showed that unlike neutrophils and lymphocytes (that express the decoy TRAIL receptors: TRAILR3), monocytes and macrophages express the functional TRAIL receptors 1 and 2: TRAILR1 and TRAILR2, and are, therefore, susceptible to the cytotoxic effect of trabectedin [101].

A preclinical study recently showed in a chronic lymphocytic leukemia mouse model that trabectedin induced leukemic cell death and depletion of the immuno-suppressive myeloid-derived suppressor cells and TAMs [102]. Anti-tumor activity by trabectedin, through effects on TAMs, has also been reported in pre-clinical models of skeletal metastatic prostate tumor, melanoma, and pancreatic tumor [103,104,105].

As macrophages play a crucial role in host defense, homeostasis, and erythropoiesis, non-specific depletion of TAMs may be harmful [106]. The scavenging receptor CD163, a marker of M2 macrophages and TAMs, has been shown to promote their pro-tumor activities in mice and humans [107,108].

Using CD163 mAbs conjugated with lipid NPs loaded with doxorubicin (DOX), Etzerodt et al. showed in a mouse melanoma model that selective depletion of CD163+ macrophages re-educates the TME through increased recruitment of T cells and monocytes, which both contribute to tumor regression. Interestingly, in this model, the pan-targeting of TAMs abrogated the therapeutic effects observed with the specific targeting of CD163+ TAMs [109]. The overall lack of efficacy observed with therapeutic approaches currently in clinical trials, such as CSF1/CSF1R or CCL2/CCR2 blockade, that indiscriminately target all macrophages, may be in part explained by these findings. A better understanding of the specific TAM subset, which mostly contributes to tumor progression, is needed.

Long-lasting macrophage depletion may also have detrimental effects. A few studies reported that prolonged CSF-1R inhibition leads to an acquired resistance and tumor recurrence through activation of the PI3K pathway; the combination of PI3K blockade with CSF-1R inhibition prolonged survival in pre-clinical models [110]. Compensatory effects and enhanced tumor progression in response to CSF-1R inhibition has been recently attributed to increased granulocyte recruitment into tumors. Speicher et al. reported that CSF-1R inhibition combined with CXCR2 antagonists blocks granulocyte infiltration and results in stronger anti-tumor effects [111].

Overall, lack of selectivity towards the specific myeloid subsets with pro-tumor activity, and problems associated with the unwanted depletion of macrophages in other organs for a prolonged period, has limited the clinical translation of these depleting strategies. Even more, recent studies indicate that reprogramming of macrophages—rather than TAM depletion—might be more beneficial in eliciting an effective anti-tumor immune response.

### 4.2. Strategies to Reprogram TAMs

Myeloid functional plasticity, as mentioned above, opens possibilities for the pharmacological reprogramming of macrophages in the light of therapeutic exploitation. Several approaches have been attempted with the aim to switch M2-like macrophages with pro-tumor properties into anti-tumor M1-like macrophages. These include the use of Toll-like receptor agonists, nucleic acids (i.e., miRNA or siRNA), and monoclonal antibodies.

#### 4.2.1. Targeting the Toll-Like Receptors for TAM Reprogramming

Toll-like receptors (TLRs) are innate immunity pattern recognition receptors that, upon engagement by their ligands, stimulate macrophages and activate an M1-like functional polarization [3]. For this reason, the capacity of different TLRs agonists to reprogram TAMs into anti-tumor effectors has been evaluated. In the last years, the pharmacological targeting of TLR3, TLR7, TRL8, and TLR9 located in the endosomal compartment of antigen-presenting cells, such as macrophages, has been prioritized versus extracellular TLRs, presumably due to their higher capacity to trigger anti-tumor immune responses. Currently, only imiquimod (TLR7 agonist) is FDA approved for topical administration in squamous and basal cell carcinoma.

We have recently shown that stimulation of tumor-conditioned macrophages with poly I:C (TLR3 agonist) is superior to imiquimod (R837) for their reprogramming into cytotoxic effectors [112]. Indeed, a crucial role for TLR3 stimulation to revert M2-macrophages polarization towards M1 has also been reported by others [113,114]. In a recent study, ROS-inducing polypeptide micelles were loaded with poly I:C and grafted with galactose groups to enable their specific uptake by TAMs in vitro and in vivo. These NPs were recognized by the galactose-specific C-type lectin (MGL) receptor expressed on the TAM surface and induced their re-education towards M1-like macrophages, activating NK and effector T cells, and ultimately leading to tumor regression in a murine melanoma model [115]. Zhao et al. reported a synergistic effect of poly I:C with Ferumoxytol^®^, an FDA-approved NP for the treatment of iron deficiency, to promote TAM polarization [116]. Combined into a unique NP, in vitro, it upregulated TNF-α and iNOS expression, increased NO secretion, and phagocytosis. In vivo, this treatment induced macrophage activation accompanied by primary and metastatic regression in a murine model of melanoma [117].

Poly I:C is being evaluated in clinical trials upon i.m. injection in combination with i.v. infusion of anti-PD-1 therapy in patients with hepatocellular carcinoma, while its analog, poly-ICLC, has been evaluated as a cancer vaccine to boost anti-tumor responses. A phase II clinical trial is investigating the sequential intra-tumor plus i.m. administration of poly-ICLC alone or in combination with ICB therapy in melanoma, head and neck and sarcoma cancers. Similarly, the TLR9 agonist CpG is currently being evaluated in clinical trials for lymphomas in combination with Ibrutinib or radiation therapy, and for the treatment of HCC in combination with the anti-OX40 mAb (BMS-986178).

Resiquimod (R848), an agonist to TLR7/8, has attracted much attention in the past few years for its capacity to reprogram macrophages. R848 is an imidazoquinoline, analog to Imiquimod, but presumably more powerful as TLR agonist, with the ability to trigger stronger anti-tumor responses [118,119,120]. Although, to our knowledge, resiquimod is not being investigated in clinical trials, several studies have been performed in melanoma patients with topical administration or local injection, mainly in combination with vaccine therapy. Despite some promising results, the systemic administration of imidazoquinolines is burdened with toxicity: whole-body inflammation, hematologic toxicity (lymphopenia, anemia), and flu-like symptoms [121,122]. To solve this problem, R848 has been covalently linked to vitamin E and modified with hyaluronic acid, thus becoming a pro-drug nanoformulation. The subcutaneous injection of this nanotherapeutic formed a depot that provided a sustained release of R848.

Another formulation of R848, MEDI9197 (3M-052), was designed to be retained at the injection site, thus limiting systemic toxicity [123]. As an alternative, to allow the intravenous administration and, at the same time, to limit its systemic toxicity, Rodell et al. developed β-cyclodextrin-NPs, loaded with R848 to target TAMs in vivo. This strategy was able to induce the production of the pro-inflammatory cytokine IL-12 by TAMs in the TME. Furthermore, the combination of these NPs with anti-PD-1 antibodies re-established their anti-tumor response in a murine cancer model resistant to ICB therapy [124].

Lately, some researchers have studied the effect of combinatorial stimulation of TLRs to achieve a synergistic activation of macrophages. In 2016, Liu et al. measured cytokine secretion by macrophages exposed to poly I:C and R848, and they found that activation of the JAK-STAT pathway through TLR3 stimulation is desirable to prime macrophages for subsequent synergistic response to TLR7 [125].

Despite these promising results in vitro, the information related to the in vivo application of combinations of TLRs for the treatment of cancer is still very limited, with only a few pieces of evidence in the field of cancer vaccination for their application as adjuvants. In this regard, NPs loaded with poly I:C and R837 [126] or R848 for vaccination purposes [127], showed the ability to trigger a strong innate immune response in lymph nodes, resulting in therapeutic efficacy without the systemic release of pro-inflammatory cytokines. Poly-ICLC plus R848 have been investigated in the clinic in combination with a cancer vaccine in patients with advanced tumors expressing the NY-ESO-1 protein. Finally, TLR agonists have also been combined with immunogenic cell death (ICD) activators, such as oxaliplatin or doxorubicin [128,129]. While the TLR agonists activate innate immunity cells, ICDs trigger the release of tumor antigens: Thus, the combined treatment should result in TAM reprogramming and strong antigen-specific T cell responses against tumors.

#### 4.2.2. RNA Delivery to Reprogram TAMs

With technological advances made in oligonucleotide delivery, mRNA, siRNA, or miRNA therapies are now promising strategies to manipulate macrophages for the treatment of cancer. As an example to deliver mRNA into TAMs, charge-altering releasable transporters (CARTs) have been developed, using oligo (carbonate-b-alfa-amino ester)s, as dynamic carriers. These have the ability to complex, protect, and deliver polyanionic mRNA through a controlled degradation and facilitate the cytosolic release of functional mRNA [130]. This method has been used to deliver a combination of OX40L, CD80, and CD86 encoding mRNAs in different subcutaneous two-tumor models, where only one tumor was treated. Upon intra-/peri-tumor administration, CARTs successfully transfected tumor-infiltrating cells, including 28% of TAMs, cured the treated tumor, and induced a systemic anti-tumor immunity, as seen by the regression of the second untreated tumor [131].

In another study, Zhang et al. administered biodegradable polymeric NPs encapsulating two mRNAs and functionalized with di-mannose moieties on their surface to target and re-educate TAMs. A first mRNA encoding IRF5, a member of the interferon regulatory factor family, and a second one encoding IKKβ, a kinase that phosphorylates and activates IRF5, were able to downregulate the expression of M2 genes, such as Serpinb2 and CCL11, and to upregulate the M1 gene CCL5. In a murine ovarian tumor model, i.p. injections effectively reprogrammed TAMs towards M1-like macrophages and increased T cell and neutrophil infiltration into the tumors. Ex vivo treated-TAMs expressed increased levels of cytokines IL-12, IFN-γ, and TNF-α, while IL-6 was decreased. The same treatment was also effective in a murine model of glioma and lung metastasis [132].

Small interfering RNA (siRNA) has been used to silence the expression of genes that regulate the immuno-suppressive functions of TAMs [133,134]. Song et al. designed mannosylated dual pH-responsive NPs loaded with 2 siRNAs directed against VEGF and placental growth factor (PIGF). These two growth factors, overexpressed in breast cancer cells and TAMs, promote tumor proliferation and immunosuppression. In a murine breast cancer model, these NPs efficiently delivered the siRNAs to both TAMs and cancer cells, resulting in gene silencing and inhibition of tumor growth and lung metastasis [135].

MicroRNA (miRNA) are small non-coding RNA molecules that function in RNA silencing and post-transcriptional regulation of gene expression. Cai et al. developed lipid-coated calcium phosphonate NPs containing conjugated mannose to facilitate the delivery of miRNA-155 to reprogram TAMs towards an M1 phenotype [136]. These NPs were shielded with a pH-responsive material preventing particle uptake at physiological pH and enabling mannose exposure and uptake by TAMs in the acidic tumor microenvironment. In a mouse sarcoma model, this delivery system reduced TAM expression of IL-10, MMP9, and VEGF, and increased expression of IL-12. This effect was associated with decreased tumor growth and prolonged survival [137].

To our knowledge, no clinical trials have been initiated using RNA delivery technology to reprogram macrophages. Old clinical trials have used several approaches to transfect mRNA for dendritic cell reprogramming; some trials are currently investigating the use of “personalized” mRNA tumor vaccines encoding neo-antigens in combination with ICB therapy, mainly for melanoma, lymphoma, head and neck and breast cancer. Another Phase I/II multicenter study is using liposomes loaded with mRNA-2416, which encodes human OX40L, in combination with anti-PD-L1 therapy in patients with advanced tumors.

#### 4.2.3. Antibodies to Reprogram TAMs

The CD47–SIRPα axis is involved in the regulation of phagocytosis. CD47 is expressed by tumor cells and interacts with the signal regulatory protein-α (SIRPα, also known as SHPS1) expressed on the surface of phagocytic cells, such as macrophages and dendritic cells. This interaction results in the inhibition of phagocytosis and thus acts as a “don’t eat me” signal, important for tissue homeostasis. Substantial evidence has been provided that overexpression of CD47 by many cancer types is an important mechanism of resistance to phagocytosis [138]. The pharmacological inhibition of CD47 restored phagocytosis and killing of tumor cells by macrophages in various preclinical cancer models, resulting in effective anti-tumor immune responses [139,140,141]. Combination therapies have the potential to increase this efficacy. As an example, the reprogramming of TAMs towards an M1 phenotype has been achieved by the combination of SIRPα-blocking antibodies with CSF-1R inhibitors assembled into a unique supramolecular-system. Repolarization of TAMs was associated with increased phagocytosis of cancer cells and enhanced anti-tumor efficacy [142,143].

Promising results were obtained in patients with aggressive or indolent lymphoma with a combination of anti-CD47 mAbs and anti-CD20 (Rituximab^®^) to target B cells [144]. Based on this evidence, several clinical trials are now being performed with anti-CD47 mAbs or CD47-Fc fusion proteins for the treatment of different tumor types, also in combination with anti-PD-1 therapy (Table 2).

Antigen-presenting cells, such as macrophages and dendritic cells, express on their surface CD40, a receptor of the TNF receptor superfamily. Interaction with its ligand CD40L, mainly expressed by T cells, basophils, and mast cells, upregulates the expression of MHC molecules and secretion of pro-inflammatory cytokines, promoting T cell activation [145]. In various murine tumor models, agonistic anti-CD40 antibodies have led to the recovery of tumor immune surveillance, mediated by the reprogramming of TAMs towards M1-polarization and effective anti-tumor activity [146,147,148]. The combination of anti-CD40 plus anti-CSF-1R antibodies has been evaluated in “cold” preclinical tumor models, not responsive to ICB. This combination therapy was able to turn “cold” into “hot” tumors with an overall decrease in immuno-suppressive cells, enhanced activity of infiltrating T cells and potent anti-tumor immunity [149]. Anti-CD40 mAbs are under evaluation in combination with checkpoint immunotherapy, chemotherapy, or targeted therapies in patients with advanced solid tumors (Table 2).

#### 4.2.4. Other Strategies to Reprogram TAMs

In 2016, Kaneda et al. investigated the role of PI3Kγ in the immuno-suppressive activity of TAMs. They showed that PI3Kγ controls the switch between immune stimulation to suppression through the inhibition of the NF-κB pathway and activation of C/EBPβ. Selective inhibition of PI3Kγ reverted this effect and resulted in pro-inflammatory cytokine expression, CD8+ T cell recruitment to tumors, and tumor growth inhibition [49]. In a preclinical model of pancreatic cancer, it has been shown that interactions between B cells and FcRγ+ TAMs promote tumor progression through M2 macrophage polarization via BTK–PI3Kγ signaling. Tumor regression was achieved by administration of ibrutinib, a BTK inhibitor, or PI3Kγ inhibitors, which reprogrammed TAMs towards an M1 phenotype that fostered CD8+ T cell cytotoxicity [52].

The PI3Kδ/γ inhibitor RP6530 was also shown to switch macrophages from an immuno-suppressive M2-like phenotype to a more inflammatory M1-like state. In tumor xenografts of Hodgkin lymphoma, the compound RP6530 repolarized TAMs, inhibited tumor vasculature, and enhanced tumor regression [150].

Histone deacetylases (HDACs) are enzymes that remove the acetyl groups on histones during the process of epigenetic regulation of gene expression. A specific Class IIa HDAC inhibitor (TMP195) has been shown to induce an inflammatory state in monocytes through modification of their epigenomic profile [151]. In a mouse model of breast cancer, the administration of TMP195 induced the recruitment and differentiation of immunostimulatory CD40+ TAMs, resulting in tumor reduction. Moreover, the combination of TMP195 with chemotherapy regimens (carboplatin and paclitaxel) and immunotherapy (anti-PD1 antibodies) significantly enhanced the stability of the tumor response [152].

Interestingly, some investigations using CSF-1R or CCR2 inhibitors showed that the anti-tumor efficacy of these drugs was to reprogram TAMs, rather than macrophage depletion, as supposed previously. For example, CSF-1R or CCR2 inhibition resulted in altered infiltration of myeloid cells by affecting CD11b+ Ly6G- Ly6Cl^+^ MHCII^+^ F4/80+ macrophages in murine pancreatic cancer models. CSF-1R or CCR2 inhibition, in combination with chemotherapy, resulted in restored CD8+ T cells anti-tumor activity [153]. Interestingly, inhibition of IL-10 receptor in macrophages also reduced tumor burden in breast cancer models if combined with chemotherapy, with an equivalent effect caused by blockade of CSF-1R, in both cases associated with increased IL-12 production by intra-tumor DCs and CD8+ T cell-mediated anti-tumor activity [154]. In mouse glioma models, the CSF-1R inhibitor BLZ945 was effective in treating established tumors and increased mouse survival. Interestingly, the authors found that GM-CSF and IFNγ helped TAMs to survive CSF-1R inhibition, leading to the reprogramming of TAMs rather than their depletion [96]. PLX3397 has also been combined with valatinib (inhibitor of VEGF) and dovitinib (inhibitor of FGF receptors) to induce reprogramming of TAMs in glioma [96].

## 5. Conclusions

Although extensive knowledge is available about the characterization and roles of macrophages in solid tumors, there is still room for investigation. The most recent studies have found significant differences in macrophages in distinct tumors, and other investigations have dissected, at the molecular level, the existence of diverse TAM subsets. Following this trend, in the near future, we expect more research to understand in detail the topographical and temporal heterogeneity of specific subsets of macrophages in primary tumors and also in their metastasis. The molecular examination of the mechanisms by which macrophages influence tumor progression and hamper the response to anti-tumor therapies is also a very active field of investigation. In parallel, intense pharmacological research works on the development of strategies to target new relevant molecules are expected. Notably, the increasing importance of macrophages in cancer has prompted scientists to evaluate the interaction of each new drug, but also of “old” pharmacological molecules, on these cells. Indeed, clinical trials using immunotherapies, but also any other kind of oncological treatment, have started to more precisely evaluate treatment efficacy also in the perspective of its effect on the patient’s immune system. Altogether, this knowledge will be relevant to design new anti-tumor therapies combining TAM-targeting approaches with other treatments.

## Figures and Tables

**Figure 1 cells-09-00046-f001:**
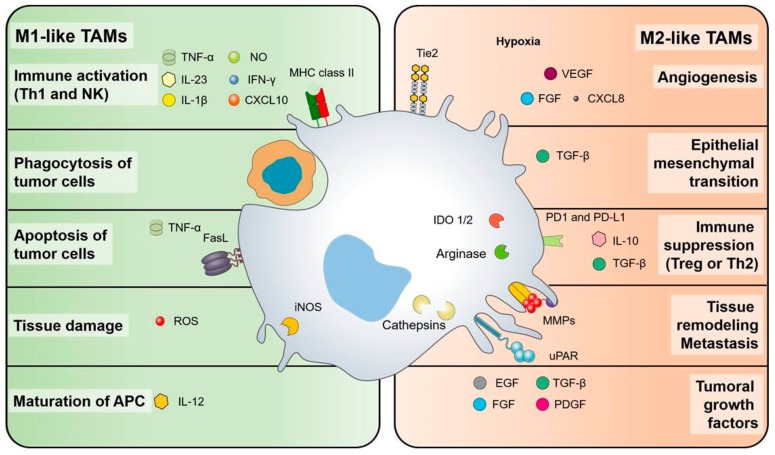
Anti-tumor and pro-tumor functions of tumor-associated macrophages (TAMs). A key feature of macrophages is their intrinsic plasticity, the two extremes of which have been identified as M1-like and M2-like polarization. In the tumor microenvironment, several molecular pathways have been recognized which drive and maintain the phenotypes and functions of TAMs. On the left side, M1-like macrophages, with anti-tumor functions, can be stimulated by immunostimulatory cytokines such as IL-1b, IL-12, IL-23, TNF-alfa, and IFNγ; MHCII molecules and IL-12 are required for efficient antigen presentation. M1-like TAMs produce chemokines, such as CXCL10, that promotes the recruitment and activation of T cells. In addition, M1-like-TAMs actively phagocytose tumor cells and release TNF-alfa, ROS, and NO for the direct killing of cancer cells. On the right, M2-like macrophages, with pro-tumor functions, are conditioned by the hypoxic tumor micro-environment and by immuno-suppressive mediators (IL-10, TGFβ). M2-like-TAMs secrete molecules to promote angiogenesis (CXCL8, VEGF), tumor proliferation (EGF, FGF, PDGF), induce epithelial-mesenchymal-transition (TGFβ), and continuous matrix remodeling (MMPs, cathepsins, uPAR). Several immuno-suppressive molecules are produced (IL-10, TGFβ, IDO1/2), which support regulatory T cells.

**Figure 2 cells-09-00046-f002:**
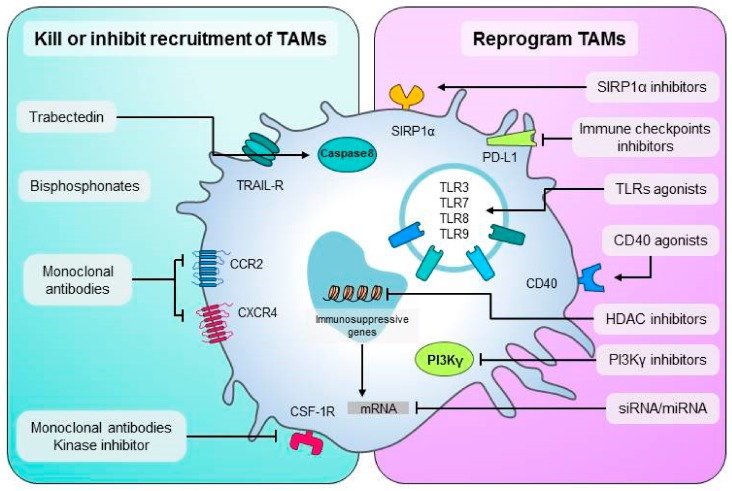
Summary of available therapeutic strategies to target TAMs. On the left side are different approaches to kill macrophages or inhibit their recruitment in tumors. Monoclonal antibodies or kinase inhibitors have been developed to disrupt the CSF-1/CSF-1R, CCL2/CCR2, or the CXCL12/CXCR4 axis required for the recruitment of new macrophages towards the tumor. Traditional bisphosphonates free or loaded into nanocarriers, and also trabectedin are chemotherapeutics, which showed preferential toxicity towards monocytes/macrophages and have been used to reduce their number in tumors. On the right side are strategies to reprogram TAMs into M1-like anti-tumor effectors. Monoclonal agonist antibodies to CD40 or agonists to Toll-like receptors activate TAMs. SIRP1α inhibitors prevent the block of phagocytosis; mAbs against immune checkpoint ligands, such as PD-L1, can also target TAMs. RNA-based therapies and some small drugs inhibiting histone acetylation (HDAC) or the PI3Kgamma pathway are also under evaluation.

**Table 1 cells-09-00046-t001:** Summary of clinical trials to inhibit the recruitment or to deplete TAMs, also in combination with other anti-tumor therapies.

CSF-1R Inhibitors (Monotherapy)				
PLX3397 (Pexidarnitib)	Melanoma	Phase II		NCT02071940
	Advanced solid tumors	Phase I		NCT02734433
	PVNS or GCT-TS	Phase III		NCT02371369
	Leukemia, sarcoma, or neurofibroma	Phase I/II		NCT02390752
	Acute myeloid leukemia	Phase I/II		NCT01349049
PLX7486 (Plexxikon)	Advanced-stage or metastatic solid tumors	Phase I		NCT01804530
		Phase I		NCT03069469
DCC-3014		Phase I		NCT01316822
ARRY-382				
LY3022855 mAb (IMC-CS4)	Solid tumors	Phase I		NCT02265536
		Phase I		NCT01346358
AMG820 mAb	Solid tumors	Phase I		NCT01444404
**CCR2 Inhibitors (Monotherapy)**				
CNTO 888 (Carlumab)	Prostate cancer	Phase II		NCT00992186
PF-04136309	Pancreatic cancer	Phase I/II		NCT02732938
MLN1202	Bone metastasis	Phase I/II		NCT01015560
**CSF-1R Inhibitors + Checkpoint Immunotherapy**				
PLX3397 (Pexidarnitib)	Pembrolizumab	Solid tumors	Phase I/II	NCT02452424
	Durvalumab	Advanced tumors	Phase I	NCT02777710
LY3022855 mAb (IMC-CS4)	Pembrolizumab	Pancreatic cancer	Phase I	NCT03153410
	Durvalumab			
	Tremelimumab	Advanced solid tumors	Phase I	NCT02718911
RO5509554/RG7155 (Emactuzumab)	Atezolizumab	Solid tumors	Phase I	NCT02323191
AMG820 mAb	Pembrolizumab	Solid tumors	Phase I/II	NCT02713529
BLZ945	PRD001	Advanced solid tumors	Phase I/II	NCT02829723
Cabiralizumab	Nivolumab	Advanced solid tumors	Phase I	NCT02526017
**CCR2 Inhibitors + Checkpoint Immunotherapy**				
BMS-813160 (CCR2/CCR5 antagonist)	Nivolumab	Advanced solid tumors	Phase I/II	NCT03184870
**CSF-1R Inhibitors + Chemotherapy**				
PLX3397 (Pexidarnitib)	Paclitaxel	Advanced solid tumors	Phase I/II	NCT01525602
	Standard Chemotherapy			NCT01042379
RO5509554/RG7155 (Emactuzumab)	Paclitaxel	Advanced solid tumors	Phase I	NCT01494688
PD-0360324 mAb	Cyclophosphamide	Ovarian cancer	Phase II	NCT02948101
**CCR2 Inhibitors + Chemotherapy**				
CNTO 888 (Carlumab)	Gemcitabine/paclitaxel	Advanced solid tumors	Phase II	NCT01204996
	Carboplatin/doxorubicin			
PF-04136309	FOLFIRINOX	Advanced solid tumors	Phase I/II	NCT01413022
**CSF-1R Inhibitors + Targeted Therapy**				
PLX3397 (Pexidarnitib)	Sirolimus (Rapamycin)	Sarcoma	Phase I/II	NCT02584647
	Eribulin	Metastatic breast cancer		NCT01596751
**CSF-1R Inhibitors + Radiotherapy**				
PLX3397 (Pexidarnitib)	RT + ADT	Prostate cancer	Phase I	NCT02472275
	RT + Temozolomide	Glioblastoma	Phase I/II	NCT01790503

**Table 2 cells-09-00046-t002:** Summary of clinical trials to reprogram TAMs, also in combination with other anti-tumor therapies.

CD47 Inhibitors (Monotherapy)				
Hu5F9-G4 mAb	Myeloid leukemia		Phase I	NCT02678338
	Acute myeloid leukemia			
CC-90002 mAb	Myeloid leukemia		Phase I	NCT02641002
	Advanced solid or hematologic cancers		Phase I	NCT02367196
SRF231 mAb	Advanced solid or hematologic cancers		Phase I	NCT03512340
TTI-621	Hematologic malignancies		Phase I	NCT02663518
(CD47-Fc fusion protein)				
**CD40 mAb Agonists (Monotherapy)**				
CP-870,893		Melanoma	Phase I	NCT02225002
**CD47 Inhibitors + Checkpoint Immunotherapy**				
TTI-621	PD-1/PD-L1 inhibitors	Relapsed and refractory solid tumors	Phase I	NCT02890368
(CD47-Fc fusion protein)				
TTI-622	PD-1 inhibitors	Relapsed and refractory lymphoma or myeloma	Phase I	NCT03530683
(CD47-Fc fusion protein)				
**CD40 mAb Agonists + Checkpoint Immunotherapy**				
APX005M	Nivolumab	Advanced solid tumors	Phase I	NCT03502330
RO7009789 (Selicrelumab)	Atezolizumab	Advanced or metastatic solid tumors	Phase I	NCT02304393
**CD47 Inhibitors + Targeted Therapy**				
Hu5F9-G4 mAb	Rituximab	Relapsed and refractory lymphoma	Phase I/II	NCT02953509
	(anti-CD20 mAb)			
**CD40 mAb Agonists + Targeted Therapy**				
RO7009789 (Selicrelumab)	Vanucizumab (anti-ANG-2-VEGF bispecific Ab)	Solid tumors	Phase I	NCT02665416
	Emactuzumab	Advanced solid tumors	Phase I	NCT02760797
	(anti-CSF-1R Ab)			

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
