# Peer review of "Current Strategies to Target Tumor-Associated-Macrophages to Improve Anti-Tumor Immune Responses"

_cells, 2019, doi:10.3390/cells9010046_

Round 1

Reviewer 1 Report

Authors present a well-written, well-documented and well-covered review on the topic of tumor-associated macrophages and their potential therapeutic targeting. Special focus is put on translational aspects and on existing knowledge on TAM-directed drugs that have been or are being tested in the clinics. Overall, this review article is very informative, well presented and deserve publication. This reviewer has a couple of suggestions that could be considered by authors to improve the manuscript.

In chapter 3 about TAMs and their interference in clinical treatments, they describe interactions of TAMs with ICB therapy, interactions of TAMs with chemotherapy and interactions of TAMs with anti-angiogenic therapies. However, they pay no attention to interactions of TAMs with radiotherapy, being the latest a mainstay of modern oncology management.
2. Tables 1, 2 and 3 in manuscripts refer to current clinical trials targeting specific pathways, namely CSF-1R, CD47 and CD40. Author could consider expanding these tables and include clinical trials targeting also other molecules. All the information given in the manuscript regarding strategies to target TAMs could be summarized in 2 tables: 1) strategies to inhibit recruitment or deplete TAMS, and 2) strategies to reprogram TAMs

Author Response

We would like to thank the Reviewer for her/his kind appreciation of our review and useful suggestions.

In the revised manuscript we have introduced a paragraph on radiotherapy and interefrence by TAMs, as well as  a couple of sentences on combination studies with radiotherapy. these sections are highlighted in red.

We have also re-named the Tables, as suggested: 1) strategies to inhibit recruitment or deplete TAMs, and 2) strategies to reprogram TAMs),

and  expanded Table 1 to include other therapeutic approaches (CCL2/CCR2 inhibition), also in red.

We feel that for other approaches there little to include in a table.

Table 2 and table 3 were merged in one table.

Reviewer 2 Report

my comment to review:Current strategies to target Tumor-Associated- Macrophages to improve anti-tumor immune responses, is  

I read with very interest the submitted manuscript: Current strategies to target Tumor-Associated- Macrophages to improve anti-tumor immune responses, written by Clément Anfray et al. that deals with the  phenotypical and functional plasticity of Tumor-Associated Macrophages (TAMs) TAM and their role in cancer progression as well in hampering  the conventional therapies, lastly  describes the therapeutic strategies targeting TAMs. The importance of the topic which is described  clearly and exhaustively makes this work suitable for publication after minor revision of the text. English language and style are fine/minor spell check required as mentioned before

Author Response

We would like to thank the Reviewer for her/his appreciation of our review.

The Whole text has been checked and several typos  have been corrected.